# Disability-adjusted life years (DALYs) based COVID-19 health impact assessment: A systematic review protocol

**Daniel Teshome Gebeyehu**[1,2]*, **Leah East**[1], **Stuart Wark**[3], **Md Shahidul Islam**[1]

**1** Faculty of Medicine and Health, School of Health, University of New England, Armidale, NSW, Australia, **2** School of Veterinary Medicine, Wollo University, Dessie, Amhara, Ethiopia, **3** Faculty of Medicine and Health, School of Rural Medicine, University of New England, Armidale, NSW, Australia

* daniel.teshome@wu.edu.et

## Abstract

### Background

COVID-19 is a highly contagious infectious disease that emerged in 2019. This disease is causing devastating health, socio-economic, and economic crises. More specifically COVID-19 is affecting both the quality and length of human life. The overall health impact of this disease is measured by the disability-adjusted life years which is the sum of the life years lost due to disability (the effect on the health quality) and the years life lost due to pre-mature death (effect on the length of life). The purpose of this review is to summarise DALYs-based health impact publications and produce compiled and informative literature that can aid the health regulators to make evidence-based decisions on mitigating COVID-19.

### Methods

The review will be conducted using the PRISMA 2020 guidelines. The DALYs-based original observational and cross-sectional studies will be collected for assessing the health impact of COVID-19. Both the life quality and length impacts of COVID-19 will be reviewed. The life quality impact of COVID-19 will be measured using the life years lost due to disability (pre-recovery illness, pre-death illness, and post-acute consequences), and its impact on the length of life will be measured with years of life lost due to premature death (shortening of life expectancy). The combined health impact of COVID-19 on the quality and length of life will be measured in disability-adjusted life years.

### Discussion

The impacts of COVID-19 on the two health outcomes (quality and length of life) will indicate the level of COVID-19 health burden. The increase or decrease of COVID-19 health impact might be due to the sample size differences of different studies and the omission of years lost due to post-acute consequences in some studies. After having a summarized systematic review health decision-makers will apply an impact-based response to COVID-19.

**Data Availability Statement:** No datasets were generated or analysed during the current study. All relevant data from this study will be made available upon study completion.

**Funding:** The author(s) received no specific funding for this work.

**Competing interests:** The authors have declared that no competing interests exist.

**Abbreviations:** AHRQ, Agency for Healthcare Research and Quality; CEA, Cost-Effectiveness Analysis; CINAHL, Cumulated Index to Nursing and Allied Health Literature; COVID-19, Coronavirus Disease of 2019; DALY, Disability-Adjusted Life Year; GRADE, Grading of Recommendations, Assessment, Development, and Evaluations; PRISMA, Preferred Reporting Items for Systematic Reviews and Meta-Analyses; PROSPERO, International Prospective Register of Systematic Reviews; ROBIS, Risk of Bias in Systematic Reviews (Healthcare); SARS-COV-2, Severe Acute Respiratory Syndrome Coronavirus Two; YLD, Years of Healthy Life Lost due to Disability; YLL, Years of Life Lost due to Premature Death.

## Trail registration

**Systematic review registration**: This protocol is pre-registered in PROSPERO with the registration number CRD42022324931.

## Introduction

Coronavirus disease (COVID-19) is an infectious disease caused by severe acute respiratory syndrome coronavirus two (SARS-COV-2). The name "COVID-19" was given by WHO on 11th February 2020. This disease is identified as a global public health threat and characterized as a pandemic on 30th January and 11th March 2020 respectively [1]. COVID-19 is causing devastating health, socio-economic, and political crises since its emergence now on [2–4]. Near half a billion cases and about 6.2 million deaths are reported, two years after its declaration as a pandemic [5]. COVID-19 is the largest public health threat since the emergence of pandemic influenza in 1918. If the mortality and morbidity rate of COVID-19 persists for a long time, it will ultimately shorten the life expectancy of humans [6]. COVID-19 has a wide variety of symptoms ranging from no symptoms to severe symptoms. According to the Australian government department of health [7], fever, cough, tiredness, and loss of taste or smell are the common symptoms that most COVID-19 patients have shown. It is expeditiously contagious and its fatality rate is variable from country to country, ranging from less than 0.1% to 25% [8]. On average, COVID-19 has 14 days of severe illness and about 28 days of post-acute consequences [8]. Regardless of their geographic location and economic status, COVID-19 is severely affecting every sector, country, and global society in general. Not only the health of patients but every individual is also affected by this disease due to lockdown, food insecurity, socio-economic disturbances, moment restrictions, and job terminations.

### Rationale

The health impact assessment using disability-adjusted life years (DALYs) is the most common quantitative health impact assessment technique [9]. One DALY represents one year lost, which is equivalent to a one-year healthy life. According to the WHO global health estimates [10], the burden of the disease for a particular cause is the sum of years of life lost due to premature mortality (YLL) and years lost due to disability (YLD). Some studies of the DALYs based COVID-19 health impact assessments had global coverages [11] and others were focused on multiple countries [12–16] while many others were targeted on specific country [17–23]. Some studies [17,18,21,22] include the post-acute consequences (tired, fatigue, and pain all over the body after acute infection) of COVID-19 in YLD calculation while others do not [13–16,18,19]. In addition, several studies didn't consider the morbidity period of patients before their death in YLD calculation [11–19] and they were simply calculated the YLL of dead patients without considering the suffering time from the onset of the diseases to death. Failing to consider both the post-acute consequences and the pre-death morbidity time, significantly reduces the DALYs due to COVID-19. The health outcome due to COVID-19 are either disturbance in quality of life (YLD) or shorten the length of life (YLL) or both of them. To assist health decision-makers in planning, prioritizing, and properly allocating health resources, it is imperative to systematically summarize the magnitude of COVID-19 health impacts in DALYs. As confirmed by a pilot search, one or two of the DALYs components (pre-death morbidity, long COVID-19/Sequelae, and suffering time from onset of infection to recovery)

are not included in previously conducted studies. As a result, publication and reporting biases are expected. Due to these expected biases, we have decided to conduct a systematic review than metanalysis.

## Objective

**Review question.** What is the impact of COVID-19 on the quality and length of human life? Based on this question, the aim of this systematic review is to show the health effect of COVID-19 by summarising DALYs-based literature and producing a representative and organized document that will aid policymakers, health managers, and health actors to make evidence-based decisions on mitigating COVID-19.

## Methodology

We have followed the PRISMA P 2020 guidelines (S 1) to prepare this protocol. In addition, we have used the Joanna Briggs Institute (JBI) reviewers' manual and Cochrane handbook for systematic reviews of interventions during the preparation of this protocol. This systematic review is pre-registered in PROSPERO with a registration number CRD42022324931.

### Eligibility criteria

The studies will be selected based on the inclusion and exclusion criteria shown in Table 1.

The DALYs-based COVID-19 health impact studies whether they consider post-acute consequence and pre-mortality illness time or not will be included and the gap of YLD calculations will be appraised according to the cost-effectiveness analysis (CEA) registry of Tufts University.

### Information sources

Literature from five databases (PubMed, Scopus, and Web of Science) will be searched from 15 May to 30 June 2022. In addition to these databases, manual searches will be applied to avoid missing valuable literature using the google search engine. During manual searching, the snowball searching technique will be applied to access related publications with our literature of interest.

### Search strategy

Different combinations of concepts will be used to address our topic of interest. Since the name of COVID-19 is variable in different literature, we will use a complex search strategy to capture valuable literature. The search term *(Impact) OR (Burden) AND ("COVID-19") OR ("COVID 19") OR ("SARS-COV-2") OR ("SARS COV 2") OR ("Coronavirus disease 2019") OR ("Coronavirus disease-19") OR ("Coronavirus diseases 19") OR ("Severe acute respiratory*

**Table 1. Inclusion and exclusion criteria for study selection.**

| No. | Inclusion criteria | Exclusion criteria |
|---|---|---|
| 1. | Studies that used DALYs metrics | Studies that didn't use DALYs metrics |
| 2. | Primary research articles | Publications other than primary articles |
| 3. | Primary research articles published between 31 December 2019 (COVID-19 emergence date) to 30 June 2022 (the end of data collection date) | Studies published before 31 December 2019 and after 30 June 2022 |
| 4. | Studies published in English language | Studies published other than English language |

*syndrome coronavirus 2") OR ("Severe acute respiratory syndrome coronavirus-2") OR ("Novel coronavirus") OR ("Wuhan coronavirus") AND (Health) AND (DALY) OR (DALYs) OR ("Disability-adjusted life years")* will be used for all databases. The proposed searching strategy for each database is attached as (S1 Text). The dates from 31 December 2019 to 30 June 2022, the English language, Observational/Cross-sectional study design, and full-length article type will be set as a filtering mechanism to generate target literature only. For manual searching, the title (*the impact of COVID-19 on human health*) will be directly written on the google search engine and the articles which are not already identified using database searching will be included.

## Selection process

The primary literature selection and screening will be done by DTG. To ensure the quality of the selected manuscripts and avoid missing important literature, the selection process will be reviewed by the three senior experts (MSI, LE, and SW) independently. If there will be inconsistencies among the reviewers, all the authors will come together and alleviate the differences with discussion or reviewing the full-text article for certainty, but if there will no consensus on the inconsistencies, the issue will be referred to an external reviewer.

## Data collection process

In the first instance, the literature will be collected by DTG using both the pre-set database searching term and manual searching. The selected online databases will be explored using EndNote X9 to identify the target literature. The whole literature generated from the databases and manual searches will be merged in EndNote X9 and deduplicated using a unique identifier in the EndNote library. After the duplicated literature is removed the less relevant literature will be avoided using the title and abstract of each article. This data collection process will be checked for accuracy by the three senior authors (MSI, LE, and SW). If there will be disagreement on the data collection process, it will be solved by discussion.

## Data items

The outcome of any literature that measures the health impact of COVID-19 in the form of DALYs will be considered. The health outcome of COVID-19 will be considered in either the length of life (YLL) or the quality of life (YLD) or both (DALY). It is normal to report the impact of COVID-19 on quality of life (YLD) if there is no death and the length of life (YLL) if there is no survival. As a result, if the studies didn't include either YLD or YLL, they will not be excluded in this regard. Ideally, the DALYs research of COVID-19 should consider years lost due to illness/disability (time from the onset of diseases until recovery, pre-mortality time, and post-acute consequences/sequelae), and the years lost due to premature mortality. Whether the literature includes or misses one or more of these components, the literature will not be excluded on this base.

Any literature regardless of its study area (global, country, city, or hospital-based) will be considered. Even though the magnitude of the COVID-19 health impact is variable based on the standards used during the study, any literature that used the global burden of diseases 2010 standard (using age-weighting, time discounting, and incidence) or that used the revised WHO global health estimate (not using age-weighting and time discounting and using prevalence instead of incidence) will be included. The number of participants and the duration of data collection will not be considered.

## Study risk of bias assessment

The risk of bias will be assessed using the ROBIS tool for systematic review [24]. It has three phases (assess relevance (optional), identify concerns with the review process, and judge the risk of bias in the review). The risk of bias will be assessed using the questions under each domain of phase 2. Each question has *"yes, probably yes, probably no, no, and no information"* alternatives/choices. Based on the answers to the questions in phase 2, the risk will be measured into *"low or high or unclear"* in phase 3. The risk of bias assessment will be performed by DGT and independently checked by the remaining three protocol authors (MSI, LE, and SW). Any discrepancies aroused among the three protocol authors will be discussed to reach on satisfactory consensus. When the data in the literature is found debatable, the original literature author will be contacted for further elaboration.

## Effect measures

The effect of COVID-19 on the quality of life (YLD) and the length of life (YLL) will be measured in the number of years lost. The DALYs as a result of COVID-19 will be analyzed using descriptive statics (percentage, range, and frequency). The magnitude of each DALY subset (years lost due to pre-recovery illness, post-acute consequence, pre-mortality illness, and premature death) will be expressed in percentages. This expression eases the prioritization and engagement of COVID-19 prevention interventions. The subset that highly contributed to the DALYs due to COVID-19 will be prioritized for immediate health intervention. The patients' demographic factors like age will be considered and the age ranges that are mostly affected by COVID-19 will be identified.

## Synthesis methods

By narrowing the multi-sectoral impacts of COVID-19, we planned to concentrate on its health impacts only. Using the two health outcomes (reducing the quality of life and the length of life) of COVID-19, the synthesis will be made according to the following questions:

A. How much the quality of life is affected by COVID-19? This question will be elaborated on using the YLD of COVID-19 and measures the compromised life of humans due to COVID-19 illness. The YLD of COVID-19 will be synthesized using three exclusive questions: A1) how much the COVID-19 patients were suffered due to pre-recovery illness, A2) how much the patients were suffered due to post-acute consequences (tired, fatigue, and pain all over the body), and A3) how much the patients were suffered due to COVID-19 before they died? All these three subsets of YLD will measure the effect of COVID-19 on the quality of life using different disability weights.

B. How much the length of life is affected by COVID-19? This question will measure the effect of COVID-19 in shortening the life expectancy (YLL) of humans.

The details of the literature (authors, study area, number of participants, life quality effect, and life length effect) will be tabulated and a pie chart will be used to show the contribution of each YLL and YLD to the cumulative DALY of COVID-19.

## Reporting bias assessment

The AHRQ tool for evaluating the risk of reporting bias in systematic reviews will be used. Find more information about the AHRQ tool here. Using the checklists in this tool [25], all protocol authors will independently assess the risk of bias due to unreported or missed results. Any likely disagreements will be solved by discussion.

### Certainty assessment

We will use the GRADE tool to assess the certainty of evidence. The factors we will consider during the certainty assessments are the directness of findings, inconsistency of findings, publication bias, study limitations, and importance of outcomes. Based on the criteria (a large effect, confounding effect, and miss calculation effect) we had set for GRADE domains, the certainty of evidence will be judged as high, moderate, low, and very low. All the protocol authors will independently conduct the certainty assessment and any discrepancies will be discussed and solved. The summary of findings tables will be prepared using the GRADEpro GDT software.

## Discussion

To the best of our knowledge, this systematic review on DALYs-based COVID-19 health impact or its similar is not done yet. Apart from its effect on the socio-economic and political sectors, COVID-19 is causing devastating national, regional, and global health crises. The study of the COVID-19 health impacts is a foundation for evidence-based decision-making of health policymakers, impute suppliers, and governmental and non-governmental organizations. COVID-19 affects both the quality and length of life.

Patients with COVID-19 are suffered due to fever /chills, cough, shortness of breath or difficulty of breathing, fatigue, muscle or body aches, headache, loss of taste or smell, and sore throat. At the time of such symptoms, patients cannot perform their healthy life and live as disabled. The measure of years lost due to disability (YLD) indicates the impact of COVID-19 on the quality of health. In addition to, the life lost due to acute illness, YLD will include the years lost due to post-acute consequences and the suffering time of patients before their death.

Not only the quality of life, but it also has a huge impact on the length of human life. People are dying due to the pandemic COVID-19. If they were not dead as a result of this disease, they expect to live more than their age of premature death due to COVID-19. COVID-19 is considerably decreasing the life expectancy of humans. The years of life lost (YLL) due to premature death indicates the impact of COVID-19 in shortening the length of life.

The general impact of COVID-19 on both health outcomes (quality and length of health), will be measured in the disability-adjusted life years (DALYs). The health burden of COVID-19 might be variable from one literature to another because some literature missing to include the post-acute consequences of COVID-19 and others include it in the YLD calculation.

Based on the above elaborations, health policymakers and decision-makers may prioritize intervention options, either intervening in the health quality improvement or life length increment. In addition, health governors may highlight the burden of COVID-19 with other types of diseases. Any diseases control, prevention, and eradication activities are depending on the impact of that disease. As a result, this review will be conducted to supply reliable information for the impact-based responses to COVID-19. The publication of this protocol can aid other protocol authors who are interested to review the impact of any disease on the quality and length of life.

### Expected limitations of included studies and data identification process

This systematic review has limitations in including all research articles published in different languages than English. The multisectoral impacts of COVID-19 other than its health burden are not included in this review. Not only this, the review didn't include the qualitative health impact assessments and focused on only observational quantitative studies that were done using DALYs based health impact metrics. The inclusion of only primary research articles and excluding grey literatures, reviews, short communications and unpublished articles might be a

source of missing valuable data. The studies for this systematic review will be identified use title and abstract screening method, which leads us to miss some important literature. A recent study confirmed that abstract screening misses around 13% of valuable studies [26].

As understood from the pilot data search, the DALYs based COVID-19 health impact studies covered more than one country. As a result, the COVID-19 health impact of one country might be reported by more than one study that leads us to over COVID-19 health impact estimations. In the contrary, the studies failing to include all DALYs based COVID-19 health impact components (pre-death morbidity, long COVID-19/Sequelae, and suffering time from onset of infection to recovery) lead us to under COVID-19 health impact estimations.

## Publication status

This protocol is prepared for the ongoing systematic review of the DALYs-based COVID-19 health impact. Any change to this protocol will be updated on the PROSPERO registration database (registration number CRD42022324931) and the updated part of the protocol will be published together with the full systematic review.

## Supporting information

**S1 Table. PRISMA-P checklists.pdf (attached as supporting information).**
(PDF)

**S1 Text. The searching strategy for each database (attached as supporting information).**
(PDF)

## Author Contributions

**Conceptualization:** Daniel Teshome Gebeyehu.

**Methodology:** Daniel Teshome Gebeyehu.

**Supervision:** Leah East, Stuart Wark, Md Shahidul Islam.

**Validation:** Leah East, Stuart Wark, Md Shahidul Islam.

**Visualization:** Leah East, Stuart Wark, Md Shahidul Islam.

**Writing – original draft:** Daniel Teshome Gebeyehu.

**Writing – review & editing:** Daniel Teshome Gebeyehu, Leah East, Stuart Wark, Md Shahidul Islam.

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
