## [Decision Letter · Decision Letter 0]

4 Aug 2022

PONE-D-22-15834

Disability-adjusted life years (DALYs) based COVID-19 health impact assessment: A systematic review protocol

PLOS ONE

Dear Dr. Gebeyehu,

Thank you for submitting your manuscript to PLOS ONE. After careful consideration, we feel that it has merit but does not fully meet PLOS ONE’s publication criteria as it currently stands. Therefore, we invite you to submit a revised version of the manuscript that addresses the points raised during the review process.

add more literature about health impact assessment using DALYsexpand protocol limitationsadd and revise literature, whenever possible, on quantitative analysis.Please, see detailed comments by reviewers. 

We look forward to receiving your revised manuscript.

Kind regards,

Bernardo Lanza Queiroz, Ph.D

Academic Editor

PLOS ONE

Journal Requirements:

2. We note that this manuscript is a systematic review or meta-analysis; our author guidelines therefore require that you use PRISMA guidance to help improve reporting quality of this type of study. Please upload copies of the completed PRISMA checklist as Supporting Information with a file name “PRISMA checklist”.

Reviewers' comments:

Reviewer's Responses to Questions

**Comments to the Author**

1. Does the manuscript provide a valid rationale for the proposed study, with clearly identified and justified research questions?

Reviewer #1: Yes

Reviewer #2: Yes

2. Is the protocol technically sound and planned in a manner that will lead to a meaningful outcome and allow testing the stated hypotheses?

Reviewer #1: Yes

Reviewer #2: Partly

3. Is the methodology feasible and described in sufficient detail to allow the work to be replicable?

Reviewer #1: Yes

Reviewer #2: Yes

4. Have the authors described where all data underlying the findings will be made available when the study is complete?

Reviewer #1: Yes

Reviewer #2: Yes

5. Is the manuscript presented in an intelligible fashion and written in standard English?

Reviewer #1: Yes

Reviewer #2: Yes

6. Review Comments to the Author

You may also provide optional suggestions and comments to authors that they might find helpful in planning their study.

Reviewer #1: I liked the protocol the way it stands. It is hard to make changes because it is a novel idea from the authors. I believe it is quite interesting

Reviewer #2: The protocol covers an interesting and really actual topic regarding the review of DALYs-based health impact publications. Although the manuscript is almost technically sounding, there is still room to be improved. First of all, the rationale section (lines 65-69) must be deepened adding more literature about the health impact assessment using DALYs. Thus, it should be useful to add some scientific evidence for each geographical macro-area to provide an overall state of the art and to better support the rationale of the study. Few examples are:

“Gianino MM et al. Burden of COVID-19: disability-adjusted life years (DALYs) across 16 European countries. European Review for Medical and Pharmacological Sciences. 2021 Sep 1;25(17):5529-41.”

“Zhao J, et al. Disease burden attributable to the first wave of COVID-19 in China and the effect of timing on the cost-effectiveness of movement restriction policies. Value Health. 2021; in press. doi: 10.1016/j.jval.2020.12.009.”

“Pifarré i Arolas H, et al. Years of life lost to COVID-19 in 81 countries. Sci Rep. 2021;11:3504. doi: 10.1038/s41598-021-83040-3”

Moreover, authors should rephrase the inclusion and exclusion criteria in the eligibility criteria subsection. In particular, the exclusion criteria are not the exact contrary of the inclusion ones. Authors could also expand their search strategy to additional scientific databases as CINAHL and EMBASE.

In addition to the search terms, it is useful to add, as a supplementary files, the whole search string, one for each database.

The present reviewer does not know if the authors already thought about the provision of a quantitative synthesis of the results (i.e., meta-analysis). It would be interesting to read the results of a meta-analysis of the publications reporting health impact assessment based on DALYs. From a technical point of view, it seems feasible given the presence of the common indicator (i.e., DALY) across the selected studies and the statistical properties that a continuous variable has. Lines 174-176 should be moved in the discussion section.

Authors should consider to expand the discussion of the protocol’s limitations.

7. PLOS authors have the option to publish the peer review history of their article (what does this mean?). If published, this will include your full peer review and any attached files.

Reviewer #1: No

Reviewer #2: No

---

## [Author Response · Author response to Decision Letter 0]

11 Aug 2022

Dear editor and reviewers 

First and foremost, we want to acknowledge the substantial time you spent reviewing our manuscript entitled “Disability-adjusted life years (DALYs) based COVID-19 health impact assessment: A systematic review protocol”. The changes are indicated in Microsoft track changes and attached as “marked manuscript”. Revised and “unmarked manuscript” is also attached separately. 

Editor’s comment: Thank you for submitting your manuscript to PLOS ONE. After careful consideration, we feel that it has merit but does not fully meet PLOS ONE’s publication criteria as it currently stands. Therefore, we invite you to submit a revised version of the manuscript that addresses the points raised during the review process.

1. Add more literature about health impact assessment using DALYs: As per the comment, 10 new references are added in the revised manuscript. 

2. Expand protocol limitations: The limitations are expanded based on the reviewer 2’s comment and indicated in the Microsoft track change.

3. Add and revise literature, whenever possible, on quantitative analysis: Our preference to conduct a systematic review than metanalysis is indicated in the rationale sections of the revised protocol. 

4. Please, see detailed comments by reviewers: All reviewer comments are addressed as follows. 

Reviewer #1:

I liked the protocol the way it stands. It is hard to make changes because it is a novel idea from the authors. I believe it is quite interesting.

Response: Dear reviewer, we want to extend our appreciation for your valuable comment. What you said above boosted our energy to continue our innovative work. Thanks once again. 

Reviewer #2: 

The protocol covers an interesting and really actual topic regarding the review of DALYs-based health impact publications. Although the manuscript is almost technically sounding, there is still room to be improved. 

1. First of all, the rationale section (lines 65-69) must be deepened adding more literature about the health impact assessment using DALYs. Thus, it should be useful to add some scientific evidence for each geographical macro-area to provide an overall state of the art and to better support the rationale of the study. 

Few examples are:

• “Gianino MM et al. Burden of COVID-19: disability-adjusted life years (DALYs) across 16 European countries. European Review for Medical and Pharmacological Sciences. 2021 Sep 1;25(17):5529-41.”

• “Zhao J, et al. Disease burden attributable to the first wave of COVID-19 in China and the effect of timing on the cost-effectiveness of movement restriction policies. Value Health. 2021; in press. doi: 10.1016/j.jval.2020.12.009.”

• “Pifarré i Arolas H, et al. Years of life lost to COVID-19 in 81 countries. Sci Rep. 2021;11:3504. doi: 10.1038/s41598-021-83040-3”

Response: Dear reviewer thanks for your constructive comment. We added 9 new references in the rationale section and 1 reference in the limitation section and all the changes are indicated in Microsoft track changes. 

2. Moreover, authors should rephrase the inclusion and exclusion criteria in the eligibility criteria subsection. In particular, the exclusion criteria are not the exact contrary of the inclusion ones. 

Response: Dear reviewer thanks for your comment. We accept the comment and rephrased the inclusion and exclusion criteria. 

3. In addition to the search terms, it is useful to add, as a supplementary file, the whole search string, one for each database.

Response: As per the comment, the searching strategy for each database is attached as supporting file (S 2).

4. The present reviewer does not know if the authors already thought about the provision of a quantitative synthesis of the results (i.e., meta-analysis). It would be interesting to read the results of a meta-analysis of the publications reporting health impact assessment based on DALYs. From a technical point of view, it seems feasible given the presence of the common indicator (i.e., DALY) across the selected studies and the statistical properties that a continuous variable has.

Response: Dear reviewer thanks for your genuine and valuable comment. The reason why we prefer to conduct a systematic review than metanalysis is indicated in the rationale section of the revised manuscript. In short, as confirmed from pilot search, the expected publication and reporting biases of the studies makes as to choose systematic reviews than metanalysis. 

5. Lines 174-176 should be moved in the discussion section.

Response: Done as per the comment. 

6. Authors should consider to expand the discussion of the protocol’s limitations.

Response: Done as per the comment.

---

## [Editor Report · Decision Letter 1]

30 Aug 2022

Disability-adjusted life years (DALYs) based COVID-19 health impact assessment: A systematic review protocol

PONE-D-22-15834R1

Dear Dr. Gebeyehu,

We’re pleased to inform you that your manuscript has been judged scientifically suitable for publication and will be formally accepted for publication once it meets all outstanding technical requirements.

Kind regards,

Bernardo Lanza Queiroz, Ph.D

Academic Editor

PLOS ONE
---

## [Editor Report · Acceptance letter]

2 Sep 2022

PONE-D-22-15834R1 

Disability-adjusted life years (DALYs) based COVID-19 health impact assessment: A systematic review protocol 

Dear Dr. Gebeyehu:

I'm pleased to inform you that your manuscript has been deemed suitable for publication in PLOS ONE. Congratulations! Your manuscript is now with our production department. 

Kind regards, 

on behalf of

Dr. Bernardo Lanza Queiroz 

Academic Editor

PLOS ONE